# SEMI-SUPERVISED DATASET CONDENSATION WITH DUAL CONSISTENCY TRAJECTORY MATCHING

## ABSTRACT

Dataset condensation synthesizes a small dataset that preserves the performance of training on the original, large-scale data. However, existing methods rely on fully labeled data, which limits their applicability in real-world scenarios where unlabeled data is abundant. To bridge this gap, we introduce a new task called **Semi-Supervised Dataset Condensation**, which condenses both labeled and unlabeled samples into a small yet informative synthetic labeled dataset, thereby enabling efficient supervised learning. We propose *Semi-Supervised Dual Consistency Trajectory Matching (SSD)*, a method that leverages semi-supervised knowledge distillation. The core of SSD is a two-stage trajectory matching framework that effectively incorporates unlabeled data. First, a teacher model is trained on the original data to generate accurate pseudo-labels using semi-supervised learning. Then, a student model is trained on the entire dataset with a novel *dual consistency regularization* loss. This loss enforces both **inter-model** consistency (between the student and teacher predictions) and **intra-model** consistency (for the student model under different input perturbations), ensuring robust performance. By aligning the training trajectories of the student model on the complete dataset and the synthetic dataset, SSD optimizes and obtains a high-quality synthetic dataset. Experiments on image classification benchmarks demonstrate that SSD consistently outperforms previous methods, achieving superior performance and efficiency in dataset condensation.

## 1 INTRODUCTION

Dataset condensation, also known as dataset distillation, aims to synthesize a compact dataset that preserves the essential knowledge of a large-scale dataset. Models trained on such synthetic datasets can achieve accuracy comparable to those trained on the original data, while offering significant benefits in terms of efficiency and privacy. In particular, synthetic datasets are highly valuable for scenarios that require repeated training on the same data, such as neural architecture search, continual learning, and knowledge distillation. Existing dataset condensation approaches typically assume that the entire dataset is fully labeled. Their core methodologies focus on aligning training dynamics between real and synthetic data, for example by matching gradients (Zhao et al., 2021), feature distributions (Zhao & Bilen, 2023), or training trajectories (Cazenavette et al., 2022). However, in real-world applications—especially in edge environments—labeled data are often scarce due to high annotation costs, while large amounts of unlabeled data are readily available. This imbalance poses a fundamental challenge: existing methods cannot be directly applied to generate high-quality synthetic datasets under semi-supervised conditions. Moreover, edge-deployed models are often required to continuously update while retaining prior knowledge, further amplifying the need for effective condensation methods that can exploit both labeled and unlabeled samples.

To tackle this *semi-supervised dataset condensation* problem, as illustrated in Figure 1, one straightforward idea is to generate pseudo-labels for the unlabeled portion of the data before applying condensation. Specifically, a semi-supervised learning method can first be used to train a model on the original dataset. This model then assigns pseudo-labels to the unlabeled samples, effectively converting the dataset into a pseudo-fully-labeled one. Existing dataset condensation techniques can subsequently be applied. However, since pseudo-labels are inevitably noisy, the quality of the resulting synthetic dataset may be degraded, leading to lower accuracy in models trained on it.

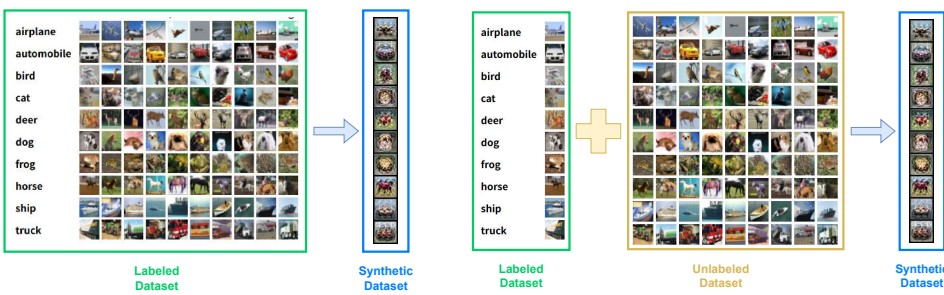

(a) Traditional Supervised Dataset Condensation    (b) Semi-Supervised Dataset Condensation

Figure 1: Difference between traditional supervised dataset condensation and semi-supervised dataset condensation. (a) Previous supervised dataset condensation methods requires all samples in the dataset are labeled. (b) Our proposed semi-supervised dataset condensation could generate high-quality synthetic dataset with **both labeled and unlabeled** samples for supervised learning.

Another naive approach builds on trajectory matching-based condensation methods. Here, a model is trained on the original dataset using semi-supervised learning, and its training trajectory is matched with that on the synthetic labeled data to produce the condensed dataset. To effectively utilize both labeled and unlabeled data, semi-supervised techniques often combine pseudo-labeling (Lee et al., 2013) with consistency regularization (Bachman et al., 2014; Sajjadi et al., 2016). For example, methods like FixMatch generate pseudo-labels from weakly augmented samples and use high-confidence ones to train on strongly augmented views (Berthelot et al., 2021). However, because the set of unlabeled samples used in each training iteration varies, trajectory matching under this setting becomes unstable and often yields suboptimal results. Alternatively, consistency-based methods such as Laine & Aila (2017); Tarvainen & Valpola (2017) minimize the prediction discrepancy between different augmentations of the same sample. Although these methods utilize all unlabeled data, they do not explicitly encourage low-entropy model outputs Grandvalet & Bengio (2004), often resulting in lower final accuracy and thereby introducing more noise during dataset condensation.

To overcome these issues, we propose a novel framework called **S**emi-**S**upervised **D**ual Consistency Trajectory Matching (**SSD**) for condensing mixed-label datasets. SSD integrates the strengths of both pseudo-labeling and consistency regularization within a knowledge distillation framework. Specifically, a teacher model is first trained on the complete dataset using semi-supervised learning to generate reliable pseudo-labels. To avoid directly using the teacher's noisy and computationally expensive training trajectory, we distill its knowledge into a compact student model. During distillation, we impose a *dual consistency regularization* loss that enforces both **inter-model** consistency (between the student and teacher predictions) and **intra-model** consistency (across different perturbations of the same input). This strategy improves the robustness and accuracy of the student model, producing high-quality training trajectories. Finally, the synthetic dataset is optimized by matching the student's training trajectories on the original and synthetic datasets. As a result, models trained solely on the small synthetic set achieve comparable accuracy on par with those trained on the full original dataset.

To sum up, our main contributions are listed below:

- We study the problem of **Semi-Supervised Dataset Condensation (SSDC)**, aiming at condense a mixture of labeled and unlabeled samples into a small, labeled synthetic dataset, thereby enabling efficient supervised learning on the small dataset. To the best of our knowedge, we are the first to study the SSDC problem.

- We propose a novel SSDC method, named by **SSD**, where we conduct semi-supervised training on the teacher model for predicting accurate pseudo-labels of unlabeled samples, while matching the training trajectories of the student model on the entire dataset with *dual consistency regularization*.

- We evaluate our method on several image classification benchmark datasets and show that SSD outperforms baselines under different settings of IPC and model architectures.

## 2 RELATED WORKS

**Dataset consensation**, also known as dataset distillation, aims to produce a compact synthetic dataset that retains essential information from the original data, such that models trained on the synthetic set achieve performance comparable to those trained on the full dataset (Wang et al., 2018; Zhao et al., 2021). To ensure the synthetic dataset captures equivalent informative content, existing methods typically optimize by matching certain signals between the original and synthetic data during training. These signals include model gradients (Zhao et al., 2021; Zhao & Bilen, 2021), feature distributions (Zhao & Bilen, 2023; Zhao et al., 2023; Zhang et al., 2024a), and training trajectories (Cazenavette et al., 2022; Liu et al., 2024; Guo et al., 2024). A common limitation across these approaches is the assumption that all original samples are fully labeled, which restricts their applicability to real-world scenarios where unlabeled data is abundant. Some recent efforts have explored dataset condensation under self-supervised learning settings, aiming to condense large unlabeled datasets into smaller unlabeled synthetic sets (Lee et al., 2024; Joshi et al., 2025; Yu et al., 2025). These methods allow models to first perform self-supervised pre-training on the condensed data before fine-tuning on downstream tasks. However, they do not incorporate label information during condensation and often require substantial computational resources. In contrast, our approach can effectively utilize available labeled samples during the condensation process, avoiding the heavy computation typical of self-supervised methods, while directly generating a semantically informative synthetic dataset that significantly reduces the cost of subsequent model training.

**Semi-supervised learning** (SSL) seeks to improve model performance by leveraging both labeled and unlabeled data. Common SSL techniques include consistency regularization (Bachman et al., 2014; Sajjadi et al., 2016), pseudo-labeling (Lee et al., 2013), and entropy minimization (Grandvalet & Bengio, 2004). Early methods such as $\Pi$-Model (Laine & Aila, 2017), MeanTeacher (Tarvainen & Valpola, 2017), and MixMatch (Berthelot et al., 2021) encourage prediction consistency under different perturbations of the same unlabeled sample. While these approaches make extensive use of unlabeled data, they do not explicitly minimize the output entropy, often leading to limited model accuracy. More recent methods like FixMatch (Sohn et al., 2020) generate pseudo-labels for unlabeled samples by retaining only those with confidence above a preset threshold. Subsequent variants such as FlexMatch (Zhang et al., 2021) and FreeMatch (Wang et al., 2023) introduce dynamic thresholding strategies—FlexMatch adjusts thresholds per class based on learning difficulty, while FreeMatch sets thresholds adaptively according to the model's global and per-class confidence. Other methods, including OTAMatch (Zhang et al., 2024b), employ optimal transport to refine pseudo-label assignment. Although these techniques improve final model accuracy by focusing on high-confidence samples, they also lead to inconsistency in the subset of unlabeled data used across training iterations. Simply combining such state-of-the-art SSL methods with dataset condensation would prevent stable utilization of the full unlabeled dataset in each training round, ultimately impairing the quality of the condensed dataset. Our proposed framework integrates the strengths of existing SSL strategies, enabling comprehensive use of unlabeled samples during condensation and improving the quality of the synthesized dataset in the semi-supervised setting.

## 3 METHODOLOGY

In this section, we first present the problem definition of semi-supervised dataset condensation, followed by the detaied description of our proposed SSD framework. The overall framework of SSD is shown in Figure 2 and Algorithm 1.

### 3.1 SEMI-SUPERVISED DATASET CONDENSATION

**Problem Definition** We consider a training dataset $D = D^l \cup D^u$, which consists of a labeled dataset $D^l = \{(x_i, y_i)\}_{i=1}^{N}$ and an unlabeled dataset $D^u = \{x_j\}_{j=N+1}^{N+M}$. Each labeled sample $(x_i, y_i) \in \mathcal{X} \times \mathcal{Y}$ denotes the $i$-th labeled image $x_i$ and its corresponding label $y_i \in \mathcal{Y} = \{1, \ldots, C\}$. Each unlabeled sample $x_j \in \mathcal{X}$ represents the $j$-th unlabeled image in $D^u$. We assume that the true

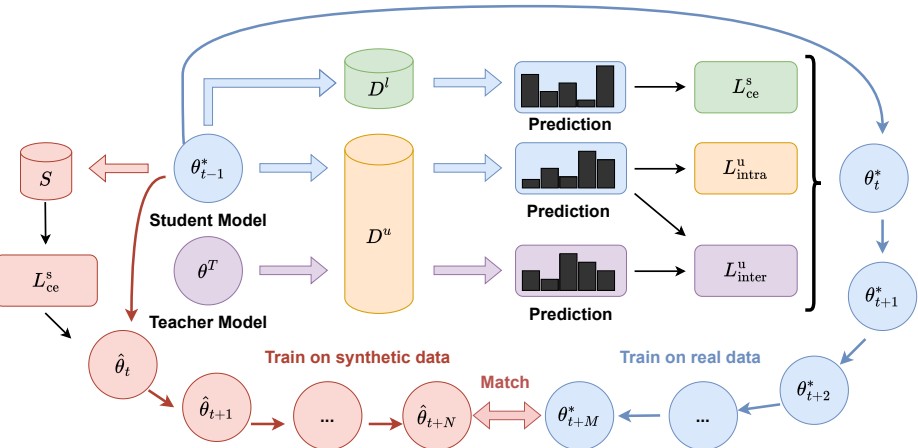

Figure 2: Overview of the SSD framework. The student model is trained with dual consistency regularization (inter-model and intra-model) to produce high-quality trajectories, which are then used to optimize the synthetic dataset by matching trajectories on the original data.

labels of the unlabeled dataset are drawn from the same label set as the labeled dataset. The test dataset is denoted as $T = \{(x_i, y_i)\}_{i=1}^{|T|}$, which is independently and identically distributed (i.i.d.) with respect to the training dataset $D$. The objective of this study is to generate a class-balanced compact dataset $S = \{(\widehat{x}_i, \widehat{y}_i)\}_{i=1}^{|S|}$, where the number of samples is significantly smaller than the original dataset, i.e., $|S| \ll N + M$. A model trained on the distilled dataset $S$ should achieve optimal performance on the test set $T$.

**Matching Training Trajectory in Supervised Dataset Condensation** Matching Training Trajectory (MTT) (Cazenavette et al., 2022) is one of the most effective strategies in traditional supervised dataset condensation, which optimizes the synthetic dataset $S$ using the following pipeline:

1. Train a model $M^D$ on the original dataset $D$ until it achieves satisfactory performance, while recording the parameters during the training process as its training trajectory.

2. Initialize $S$ either by randomly sampling images from $D$ or by using Gaussian noise, and assign labels to $S$ with predictions from $M^D$.

3. Initialize the student model $M^S$ with the model $M^D$'s parameters $\theta_t$ from the $t$-th epoch, following the training on the synthetic dataset $S$ for $N$ epochs to obtain the parameters $\hat{\theta}_{t+N}$.

4. Optimize the synthetic dataset $S$ by minimizing the loss

$$L_{\text{match}}\left(\hat{\theta}_{t+N}, \theta_{t+M}^*, \theta_t^*\right) = \frac{\|\hat{\theta}_{t+N} - \theta_{t+M}^*\|_2^2}{\|\theta_t^* - \theta_{t+M}^*\|_2^2},$$

where $\theta_t^*$ and $\theta_{t+M}^*$ denote the parameters of the model $M^D$ after the $t$-th and $(t + M)$-th training epochs, respectively, and $\hat{\theta}_{t+N}$ denotes the parameters of the student model $M^S$ obtained through inner-loop optimization using cross-entropy loss.

**Motivation** In supervised dataset condensation, training trajectories are generated by optimizing all samples in the original dataset. However, in semi-supervised learning, to improve the overall accuracy of the model, only unlabeled samples for which the model is sufficiently confident in classification are used to train the model's classification consistency. That is, the unlabeled samples used during training are dynamically changing, which leads to unstable trajectory matching and thus prevents the generation of high-quality synthetic datasets. Therefore, existing dataset condensation methods cannot be directly combined with traditional semi-supervised learning methods. There is a need to redesign new strategies to fully utilize unlabeled data for dataset condensation.

---

**Algorithm 1** SSD: semi-supervised dataset condensation with dual consistency trajectories

---

**Input:** Labeled dataset $D^l$, unlabeled dataset $D^u$, the number of student models $K$, length of trajectories $M$, learning rate $\alpha_0$ for student training, the number of iteration for outer loop $I_o$ and inner loop $I_i$.

**Output:** Synthetic data $S$ and learning rate $\alpha$

1: Train a teacher model with semi-supervised learning objective $L_{\text{pre}}$ on dataset $D^l$ and $D^u$.
2: Train and record training trajectories $\{\tau_i^*\}$ of $K$ student models with dual consistency regularization objective $L_{\text{ref}}$ on dataset $D^l$ and $D^u$ with learning rate $\alpha_0$.
3: Initialize synthetic data $S = \{(\widehat{x_i}, \widehat{y_i})\}_{i=1}^{|S|}$ and learnable learning rate $\alpha = \alpha_0$.
4: **for iteration** $i = 1$ **to** $I_o$ **do**
5:      Sample a student training trajectory $\tau^* \sim \{\tau_i^*\}$ with $\tau^* = \{\theta_t^*\}_0^T$.
6:      Choose a random start epoch $t \leq T^+$.
7:      Initialize a reference model with student params $\hat{\theta}_t = \theta_t^*$.
8:      **for iteration** $j = 1$ **to** $I_i$ **do**
9:          Sample a mini-batch of synthetic images $b_{t+j} \sim T$.
10:          Update reference model with $\hat{\theta}_{t+j+1} = \hat{\theta}_{t+j} - \alpha \nabla L_{\text{ce}}^{\text{s}}(b_{t+j})$.
11:      **end for**
12:      Compute loss $L_{\text{match}}\left(\hat{\theta}_{t+I_i}, \theta_{t+M}^*, \theta_t^*\right)$ with the ending of student and reference params.
13:      Update $S$ and $\alpha$ with respect to $L_{\text{match}}$.
14: **end for**
15: **return** synthetic dataset $S$ and learning rate $\alpha$

---

## 3.2 Semi-Supervised Dual Consistency Trajectory Matching

**Teacher Model Training** To generate a synthetic dataset that adequately represents the entire original dataset, traditional supervised dataset condensation requires using all samples of each category in the original dataset to train the synthetic dataset. However, in semi-supervised learning scenarios, some samples are unlabeled, making it impossible to directly apply existing methods for dataset condensation. A straightforward solution, therefore, is to generate pseudo-labels for unlabeled samples via semi-supervised learning, thereby enabling direct application of dataset condensation methods for training. To generate high-quality pseudo-labels, an teacher model needs to be trained on the original dataset using semi-supervised learning. A common strategy in semi-supervised learning involves leveraging high-confidence pseudo-labels from weakly-augmented unlabeled data to supervise the model's predictions on their strongly-augmented versions, thereby enforcing consistency regularization (Sohn et al., 2020; Xu et al., 2021; Wang et al., 2023). For example, the loss function of FixMatch (Sohn et al., 2020) is given by

$$L_{\text{pre}} = L_{\text{ce}}^{\text{s}} + \lambda_{\text{fix}}^{\text{u}} L_{\text{fix}}^{\text{u}},$$

where $L_{\text{ce}}^{\text{s}}$ is the cross-entropy loss on the labeled dataset, and

$$L_{\text{fix}}^{\text{u}} = \sum_{j=N+1}^{N+M} \mathbb{I}\left(\max\left(p_T\left(\alpha\left(x_j\right)\right)\right) \geq \gamma\right) H\left(\widehat{y}_j, p_T\left(\mathcal{A}\left(x_j\right)\right)\right)$$

is the consistency regularization loss on the unlabeled dataset, $H(\cdot, \cdot)$ is the cross-entropy loss function, $p_T(x)$ is the probability output by the teacher model after predicting sample $x$, $\hat{y}_j = \arg\max p_T\left(\alpha\left(x_j\right)\right)$ is the pseudo-label generated for the unlabeled sample $x_j$, $\gamma$ is the confidence threshold used to filter pseudo-labels, and $\alpha(\cdot)$ and $\mathcal{A}(\cdot)$ represents a weak and a strong data augmentation function, respectively. After training this model, all unlabeled samples can be assigned corresponding pseudo-labels, which provide category information of unlabeled samples for dataset condensation. However, since these pseudo-labels may contain noise, directly using them for dataset condensation is not an optimal choice.

**Dual Consistency Training Trajectory** To mitigate the impact of noise in pseudo-labels, we proposes dual consistent training trajectory matching to train the synthetic dataset. Trajectory matching requires obtaining training trajectories of models, i.e., the parameter changes of the model after each iteration updates on the original dataset. As discussed earlier, the teacher model trained via conventional semi-supervised learning only uses unlabeled samples with high confidence during training,

meaning the samples used in each round are inconsistent. Using the trajectory of the teacher model for dataset condensation leads to unstable results. Therefore, we proposed the use of **dual consistency regularization** to generate training trajectories across the entire unlabeled dataset. For the student model that generates the training trajectories, we apply an **intra-model consistency** regularization to all unlabeled samples, that is

$$L_{\text{intra}}^{\text{u}} = \sum_{j=N+1}^{N+M} \| p_S \left( \alpha \left( x_j \right) \right) - p_S \left( \alpha \left( x_j \right) \right) \|_2^2,$$

where $p_S(x)$ is the probability output by the student model after predicting sample x. To enhance the student model's ability to recognize unlabeled samples, we apply an **inter-model consistency** regularization that requires the model's output for samples to align with the teacher model's output, i.e.

$$L_{\text{inter}}^{\text{u}} = \sum_{j=1}^{N+M} D_{\text{KL}} \left( p_T \left( \alpha \left( x_j \right) \right) \parallel p_S \left( \alpha \left( x_j \right) \right) \right).$$

Therefore, the student model can be trained on the complete unlabeled dataset through such dual consistency regularization, generating stable training trajectories. The complete loss function is as follows:

$$L_{\text{ref}} = L_{\text{ce}}^{\text{s}} + \lambda_{\text{intra}}^{\text{u}} L_{\text{intra}}^{\text{u}} + \lambda_{\text{inter}}^{\text{u}} L_{\text{inter}}^{\text{u}}.$$

By optimizing the student model using $L_{\text{ref}}$ via mini-batch gradient descent until convergence, and recording the model parameters in each round, the required training trajectory $\{\theta_i\}$ can be obtained.

After obtaining the training trajectories, we can generate a fully labeled synthetic dataset through trajectory matching. In each round of dataset distillation, we first sample a sub-trajectory of length $M$ from the training trajectories $\{\theta_i\}$, denoted as $\{\theta_t^*, \ldots, \theta_{t+M}^*\}$. Then, starting from $\theta_t^*$ as the initial parameters of the student model, we perform $N$ rounds of updates on the synthetic dataset $S$ to obtain the parameters $\hat{\theta}_{t+N}$. Note that traditional training trajectory matching methods use the same loss function to update both the student model and the reference model. However, in this case, since we aim to obtain a synthetic dataset that can be directly used for supervised learning, we only employ cross-entropy $L^S$ to update the reference model:

$$L_{\text{ce}}^{\text{s}} = \sum_{i=1}^{|S|} H \left( \hat{y}_i, p_R \left( \alpha \left( \hat{x}_i \right) \right) \right).$$

Finally, by minimizing the discrepancy between the training endpoint of the student model and the endpoint of the reference model's training trajectory, the resulting synthetic dataset can simulate the original dataset that contains both labeled and unlabeled samples.

## 4 EXPERIMENTS

In this section, we evaluate the performance of SSD on several common image classification datasets by comparing the accuracy of models trained on synthetic datasets generated by SSD versus those produced by supervised condensation methods. Subsequently, we conduct ablation studies to validate the effectiveness and stability of key components in the SSD framework.

### 4.1 EXPERIMENTAL SETUP

**Dataset** We validated SSD on three commonly used benchmark datasets for image classification, including MNIST (LeCun et al., 1998), Fashion-MNIST (Xiao et al., 2017) and CIFAR10 (Krizhevsky, 2009). For each dataset, we randomly selected 10% of the samples from each category as labeled samples, while the remaining samples served as unlabeled data.

**Baselines** Our baselines are listed as below:

- **Random**: Randomly select samples from the labeled dataset to form the balanced condensed dataset.
- **DC** (Zhao et al., 2021): The condensed dataset is obtained by performing gradient matching on the labeled dataset.

Table 1: Accuracy comparison of condensed datasets on MNIST, Fashion-MNIST, and CIFAR-10 datasets. 10% of samples are labeled. We generate 1, 10 and 50 images per classes (IPC) for each dataset, respectively.

| Dataset | MNIST | | | Fashion-MNIST | | | CIFAR-10 | | |
|---|---|---|---|---|---|---|---|---|---|
| IPC | 1 | 10 | 50 | 1 | 10 | 50 | 1 | 10 | 50 |
| Random | 61.92 | 95.36 | 97.82 | 53.87 | 73.83 | 82.23 | 18.05 | 26.38 | 43.77 |
| DC | 90.99 | 97.10 | 98.82 | 70.10 | 82.45 | 83.52 | 27.86 | 43.63 | 51.95 |
| DSA | 85.60 | 96.25 | 98.12 | 70.72 | 82.35 | 82.97 | 27.75 | 44.82 | 51.82 |
| DM | 87.31 | 96.38 | 98.26 | 72.64 | 85.53 | 87.10 | 25.57 | 46.15 | 60.63 |
| MTT | 86.19 | 95.66 | 98.83 | 75.37 | 84.28 | 89.01 | 39.26 | 55.49 | 62.01 |
| M3D | 85.99 | 96.67 | 98.09 | 72.50 | 83.21 | 87.46 | 28.72 | 47.81 | 59.64 |
| SSD | **91.18** | **97.35** | **99.08** | **77.41** | **87.59** | **90.56** | **42.19** | **58.67** | **63.27** |

- **DSA** (Zhao & Bilen, 2021): On the basis of DC, differentiable siamese augmentation is performed on the dataset.

- **DM** (Zhao & Bilen, 2023): The condensed dataset is obtained by performing distribution matching on the samples of each category in the labeled dataset.

- **MTT** (Cazenavette et al., 2022): The condensed dataset is obtained through trajectory matching, where the training trajectories are generated via FixMatch (Sohn et al., 2020) on the entire dataset.

- **M3D** (Zhang et al., 2024a): An improved version of DM, where the condensed dataset is obtained by minimizing the Maximum Mean Discrepancy (MMD) between the representation distributions of the labeled dataset and the synthetic dataset.

**Implementation Details** Following the previous works (Zhao et al., 2021; Zhao & Bilen, 2021; Cazenavette et al., 2022), we employ a convolutional neural network (ConvNet) (Gidaris & Komodakis, 2018) as the architecture of the student model in our experiments. The ConvNet consists of three repeated convolutional blocks serving as the feature extractor. Each block comprises 128 filters, followed by average pooling, a ReLU activation function, and instance normalization. Following feature extraction, the ConvNet predicts sample categories through a linear classifier. We use a ResNet18 (He et al., 2016) as a teacher model, which is trained on the entire dataset with FixMatch (Sohn et al., 2020) objective. We adopt the same differentiable augmentation strategy following previous work (Zhao & Bilen, 2021; 2023; Cazenavette et al., 2022). We use the SGD optimizer to optimize the student model with the learning rate $\eta = 0.01$ and the batch size 256. For learning rate of synthetic images, the number of inner and outer loop iteration, and other hyperparameters, we follow the settings in previous work (Cazenavette et al., 2022).

## 4.2 EXPERIMENTAL RESULTS AND ABLATION STUDIES

**Main Results** We generated synthetic datasets on MNIST, Fashion-MNIST, and CIFAR-10, and present in Table 1 the accuracy of models trained on these synthetic sets, where IPC denotes the number of images synthesized per class. Sample synthetic images generated by SSD with IPC=10 are visualized in Figure 3. As the results demonstrate, SSD achieves the best performance across all datasets. On the relatively simple MNIST dataset, SSD outperforms the strongest baseline by a narrow yet consistent margin of 0.19%–0.26% in average accuracy under three different settings. In contrast, on the more challenging Fashion-MNIST and CIFAR-10 datasets, SSD surpasses the best baselines method by 2.04%–3.18%. Moreover, the advantage of SSD is more pronounced under low-IPC conditions, indicating its ability to produce higher-quality synthetic datasets with limited samples. These results confirm that SSD effectively addresses the task of semi-supervised dataset condensation, outperforming conventional dataset compression approaches.

**Effectiveness of semi-supervised pretraining.** To evaluate the effectiveness of using a semi-supervised pre-trained teacher model, we trained a teacher model using only cross-entropy loss without leveraging unlabeled data, and used it to guide the student model's training. The results

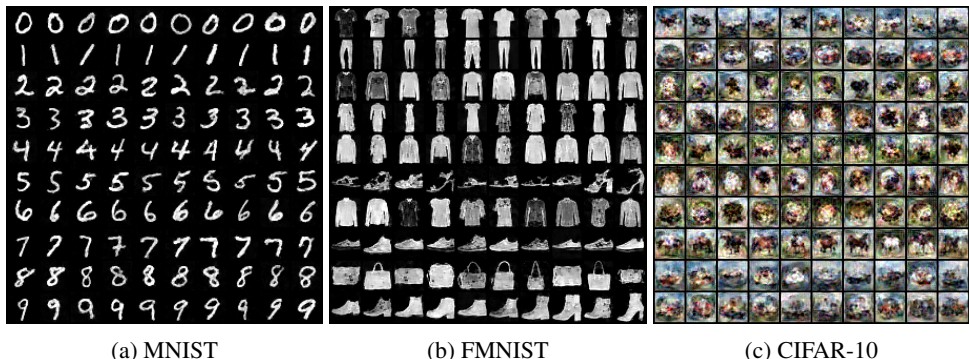

| (a) MNIST | (b) FMNIST | (c) CIFAR-10 |

Figure 3: Visualization of synthetic dataset of SSD.

Table 2: Ablation results on Fashion-MNIST and CIFAR-10 datasets.

| Dataset | Fashion-MNIST | | | CIFAR-10 | | |
|---|---|---|---|---|---|---|
| IPC | 1 | 10 | 50 | 1 | 10 | 50 |
| SSD | **77.41** | **87.59** | **90.56** | **42.19** | **58.67** | **63.27** |
| w/o Semi-Supervised Pretraining | 74.56 | 85.17 | 89.95 | 37.24 | 53.96 | 57.57 |
| w/o Inter-Model Consistency | 75.83 | 86.64 | 87.85 | 40.51 | 51.40 | 59.28 |
| w/o Intra-Model Consistency | 76.87 | 86.66 | 87.06 | 39.54 | 55.19 | 61.15 |

are presented in Table 2. Since the teacher model was trained solely on the labeled data via supervised learning and did not exploit the knowledge from unlabeled samples, it achieved relatively low accuracy. This limitation caused the student model to receive more erroneous pseudo-labels, ultimately leading to the condensation of a lower-quality synthetic dataset. The use of these inaccurate pseudo-labels resulted in significant performance degradation, with accuracy drops of up to 12.2These results clearly demonstrate the critical role of semi-supervised pre-training for the teacher model in our framework.

**Effectiveness of dual consistency regularization.** To evaluate the effectiveness of the dual consistency regularization, we conducted two ablation studies: (1) training the student model using only cross-entropy loss and intra-model consistency (applied to labeled and unlabeled data, respectively), and (2) training with only cross-entropy loss and inter-model consistency. Results are summarized in Table 2. Removing either form of consistency transforms the student training into pure semi-supervised learning or semi-supervised knowledge distillation, respectively. In the first case, the limited capacity of the compact student model makes it difficult to effectively reduce the output entropy for unlabeled samples using consistency regularization alone. In the second case, training the student with soft pseudo-labels from the teacher lacks the internal consistency enforcement against perturbed samples and becomes more vulnerable to incorrect pseudo-labels. The absence of either consistency component leads to a decline in model accuracy, thereby degrading the quality of the condensed dataset. This resulted in maximum accuracy drops of 5.2% on Fashion-MNIST and 3.2% on CIFAR-10. These results unequivocally demonstrate the importance of the proposed dual consistency regularization.

**Impact of different student and teacher model architectures.** To evaluate the impact of network architecture on the quality of the synthesized dataset, we conducted experiments by varying the architectures of both the teacher and student models within the SSD framework, with results reported in Table 3. The experimental results indicate that as the model capacity increases, the quality of the resulting synthetic dataset improves accordingly, which aligns with observations from prior supervised dataset condensation studies. These findings demonstrate the generalization ability of SSD across different network architectures.

**Impact of different semi-supervised pretraining methods.** In SSD, we obtain a teacher model through semi-supervised learning, with the expectation that it will generate pseudo-labels of suffi-

Table 3: Impact of different student and teacher model architectures on CIFAR-10.

| Teacher Arch. | Student Arch. | 1 | 10 | 50 |
|---|---|---|---|---|
| ResNet10 | ConvNet | 42.34 | 56.08 | 62.75 |
| ResNet10 | ResNet10 | 45.98 | 62.83 | 70.21 |
| ResNet18 | ConvNet | 42.19 | 58.67 | 63.27 |
| ResNet18 | ResNet10 | 46.96 | 64.86 | 71.08 |

cient quality. Therefore, the choice of semi-supervised learning algorithm may significantly impact the quality of the resulting synthetic dataset. In this section, we employ different semi-supervised learning algorithms—including Π-Model (Laine & Aila, 2017), MixMatch (Berthelot et al., 2021), FixMatch (Sohn et al., 2020), and FreeMatch (Wang et al., 2023)—to train the teacher model, investigate their effect on dataset quality, and report the results in Table 4. Although the accuracy of models trained with different methods varies to some extent, the ultimate impact on the synthetic dataset is minimal, demonstrating the robustness of SSD to the selection of the semi-supervised learning approach.

Table 4: Impact of different semi-supervised learning method for training the teacher model on CIFAR-10.

| IPC | 1 | 10 | 50 |
|---|---|---|---|
| Π-Model | 41.33 | 57.97 | 63.89 |
| MixMatch | 45.89 | 57.88 | 61.59 |
| FixMatch | 42.19 | 58.67 | 63.27 |
| FreeMatch | 46.06 | 57.34 | 65.33 |

## 5 THE USE OF LARGE LANGUAGE MODELS (LLM)

In the writing of this paper, we employed a large language model (LLM) solely for the purpose of polishing and refining the text. Specifically, we used the model to improve the fluency, clarity, and conciseness of our original drafts. All ideas, theoretical analyses, experimental designs, and results remain entirely our own. The LLM was not used to generate any scientific content or insights.

## 6 CONCLUSION

In this paper, we introduced a new task called **Semi-Supervised Dataset Condensation**, which condenses both labeled and unlabeled samples into a small yet informative synthetic labeled dataset, thereby enabling efficient supervised learning. To tackle this problem, we propose *Semi-Supervised Dual Consistency Trajectory Matching (SSD)*, a method that leverages semi-supervised knowledge distillation. The core of SSD is a two-stage trajectory matching framework that effectively incorporates unlabeled data. First, a teacher model is trained on the original data to generate accurate pseudo-labels using semi-supervised learning. Then, a student model is trained on the entire dataset with a novel *dual consistency regularization* loss, which enforces both **inter-model** consistency (between the student and teacher predictions) and **intra-model** consistency (for the student model under different input perturbations), ensuring robust performance. By aligning the training trajectories of the student model on the complete dataset and the synthetic dataset, SSD optimizes and obtains a high-quality synthetic dataset. Experiments on image classification benchmarks demonstrate that our method outperforms all baselines consistently. In the future, we plan to extend our method to more challenging scenarios, particularly to dataset condensation in environments where the dataset undergoes dynamic changes.

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
