# OpenReview forum: "Semi-Supervised Dataset Condensation with Dual Consistency Trajectory Matching"
_ICLR.cc/2026/Conference — ICLR 2026 Conference Withdrawn Submission_

### Official Review · Reviewer_fXkq · 2025-10-18

**Soundness:** 3
**Presentation:** 2
**Contribution:** 2
**Rating:** 2
**Confidence:** 4

**Summary:**

This paper  introduced a new task: Semi-Supervised Dataset Condensation (SSDC), which aims to condense a mixture of labeled and abundant unlabeled samples into a small, informative synthetic labeled dataset. To ensure robust performance and stable trajectories, student model generates training trajectories on the entire dataset using a dual consistency regularization loss. Experiments show that the proposed method consistently outperforms supervised DC baselines on MNIST, Fashion-MNIST, and CIFAR-10.

**Strengths:**

1. This work is the first one to study the SSDC problem.
2. SSD consistently and significantly outperforms several supervised DC baselines (DC, DSA, DM, MTT, M3D) across three different datasets and can be used with various SSL pre-training methods.

**Weaknesses:**

1. The novelty of the proposed method is limited. The intra-model consistency and the inter-model consistency has been used in previous SSL papers [1,2,3,4].
2. The presentation is not good. For example, there is no equation labels. In line 276, two terms are the same. Why is $L_{ref}$ used for the student network, while $L^s_{ce}$ is used for the reference model?
3. Missing latest baselines for dataset condensation, e.g. DATM [5], DANCE [6], D3S[7]. Specially,  SSDC with pseudo-labels can be seen as dataset distillation with domain shift [7].

[1] Regularization With Stochastic Transformations and Perturbations for Deep Semi-Supervised Learning. NeurIPS 2016

[2] Temporal ensembling for semi-supervised learning. ICLR 2017

[3] Mean teachers are better role models: Weight-averaged consistency targets improve semi-supervised deep learning results. NeuIPS 2017

[4] Deep Co-Training for Semi-Supervised Image Recognition. ECCV 2018

[5] Towards Lossless Dataset Distillation via Difficulty-Aligned Trajectory Matching. ICLR 2024

[6] DANCE: Dual-View Distribution Alignment for Dataset Condensation. IJCAI 2024

[7] Large Scale Dataset Distillation with Domain Shift. ICML 2024

**Questions:**

See above.

---

### Official Review · Reviewer_ptno · 2025-10-29

**Soundness:** 2
**Presentation:** 2
**Contribution:** 2
**Rating:** 2
**Confidence:** 4

**Summary:**

This paper introduces a new problem setting in the dataset condensation literature, semi-supervised dataset condensation, where a small amount of labeled data and a large amount of unlabeled data are available. Under this setup, the authors propose semi-supervised dual consistency trajectory matching, building on Matching Training Trajectories (MTT; Cazenavette et al., 2022), which updates synthetic data so that the parameter trajectory of a student model matches the training trajectory of real data (teacher model). Specifically, the method trains teacher models under semi-supervision and student models with a dual consistency loss. Experimental results demonstrate that the proposed approach achieves strong performance.

**Strengths:**

- This paper introduces a new problem setup in the dataset condensation literature.

- The proposed method is simple and easy to implement.

- The method demonstrates strong performance under the experimental settings presented in the paper.

**Weaknesses:**

- This paper lacks clarity:
    - In L270–271, could the authors clarify the statement “Using the trajectory of the teacher model for dataset condensation leads to unstable results”? Could the authors also provide empirical evidence supporting this claim?
    - In L207, what does $N$ represent in $\hat{\theta}_{t+N}$? As far as I understand, $N$ denotes the number of supervised data points, i.e., $|D^l|$.
    - In L294, the authors use $N$ to refer to rounds rather than the number of data points, which is confusing. What does a “round” mean here? Is it the number of iterations used to update the student models $p_S$?
    - In L226 (line 5 of Algorithm 1), I believe it should be $t=0$ instead of just 0.
    - In L227 (line 6 of Algorithm 1), what does $T^+$ denote?
    - In L299–301, what is $p_R$? Is it the same as $p_S$ (the student model)?

- Although this paper introduces a new problem setup, its technical contribution is limited. The main algorithm is essentially the same as MTT (Cazenavette et al., 2022), with semi-supervised teacher models.

- The motivation behind the proposed dual consistency is not persuasive. It is unclear why this component is necessary for the given problem setup.

- The experimental setup is too limited and small-scale. The baseline method, MTT (published in 2022) conducted experiments on larger datasets such as TinyImageNet (64×64) and ImageNet subsets (128×128).

- The main intuition appears to be that better teacher models under semi-supervised settings will lead to better synthetic data. If this is the case, could the authors provide the performance of the teacher models?

**Questions:**

see the weakness

---

### Official Review · Reviewer_czHj · 2025-10-31

**Soundness:** 1
**Presentation:** 2
**Contribution:** 2
**Rating:** 2
**Confidence:** 5

**Summary:**

This paper introduces Semi-Supervised Dataset Condensation (SSDC), and a method, Semi-Supervised Dual Consistency Trajectory Matching (SSD), to address it. The method employs a two-stage knowledge distillation framework. First, a teacher model is trained on the full dataset using a semi-supervised learning (SSL) technique to generate accurate pseudo-labels for the unlabeled data. Instead of matching the teacher's potentially noisy training trajectory, SSD trains a student model on the full pseudo-labeled dataset using a novel dual consistency regularization loss. This loss enforces both inter-model consistency (matching the teacher's predictions) and intra-model consistency (maintaining stable predictions under augmentation). The stable training trajectory of this student model is then used to optimize the synthetic dataset via trajectory matching. Experiments on MNIST, Fashion-MNIST, and CIFAR-10 show that SSD consistently outperforms existing supervised condensation methods adapted to this new semi-supervised setting.

**Strengths:**

The paper is easy to follow.

**Weaknesses:**

1. The novelty lies in the specific combination and application of existing techniques to the new SSDC problem. The core components—trajectory matching (MTT), semi-supervised learning with pseudo-labels (FixMatch), and consistency regularization—are all established concepts. The primary contribution is the architectural design that synthesizes these ideas into a framework that successfully stabilizes trajectory matching in a semi-supervised context. The dual consistency loss is a nice refinement but is conceptually similar to losses used in knowledge distillation and SSL. Also, the idea has alreadly been studied in previous works[1][2][3].

2. Experiments:
- Baseline Comparisons: The paper compares SSD against supervised dataset condensation methods (DC, DM, MTT, etc.) applied to the small labeled portion of the data. While SSD's superior performance is expected and demonstrates the value of using unlabeled data, a more insightful baseline would be to apply these supervised methods to a dataset where the unlabeled data has been pseudo-labeled by the teacher (the "naive approach" mentioned in the intro). This would directly isolate the benefit of the dual-consistency student trajectory from the benefit of just having more (pseudo-labeled) data. The current "MTT" baseline seems to do this, but the description is slightly ambiguous ("generated via FixMatch on the entire dataset"). Clarifying this setup is important.
- Limited Dataset Diversity: The experiments are conducted on three standard, relatively simple image classification benchmarks (MNIST, Fashion-MNIST, CIFAR-10). While sufficient for a proof of concept, the true challenge of SSL and condensation often appears in more complex, fine-grained, or long-tailed datasets, e.g., ImageNet-1K, COCO, VOC, etc. Evaluating SSD in such scenarios would provide a stronger test of its robustness.
- Computational Cost: The proposed method involves three distinct training stages: (1) training a teacher model, (2) training a student model to generate trajectories, and (3) optimizing the synthetic dataset via trajectory matching. This is a highly complex and computationally intensive pipeline. The paper lacks a clear analysis of this overhead compared to baselines. While the resulting condensed set is efficient to train on, the cost of creating it appears substantial. Also, the better the teacher, the better the performance. Please consider using weak-to-strong strategies.
- Missing baselines: Trajectory matching is a too out-dated baseline. Please conduct experiments on matching-based SOTA methods like NCFM[4], etc. Particularly, NCFM is more efficienct than MTT-based method. It could be a new direction for considering using your recipe for distribution matching. Also, decoupled-based baslines are also missing, including training-free RDED[5], and training-based SRe2L[6].

[1] Yu S F, Yao J J, Chiu W C. Boost self-supervised dataset distillation via parameterization, predefined augmentation, and approximation[C]//The Thirteenth International Conference on Learning Representations. 2025.
[2] Joshi S, Ni J, Mirzasoleiman B. Dataset Distillation via Knowledge Distillation: Towards Efficient Self-Supervised Pre-Training of Deep Networks[J]. arXiv preprint arXiv:2410.02116, 2024.
[3] Lee D B, Lee S, Ko J, et al. Self-Supervised Dataset Distillation for Transfer Learning[C]//The Twelfth International Conference on Learning Representations.
[4] Wang S, Yang Y, Liu Z, et al. Dataset distillation with neural characteristic function: A minmax perspective[C]//Proceedings of the Computer Vision and Pattern Recognition Conference. 2025: 25570-25580.
[5] Sun P, Shi B, Yu D, et al. On the diversity and realism of distilled dataset: An efficient dataset distillation paradigm[C]//Proceedings of the IEEE/CVF Conference on Computer Vision and Pattern Recognition. 2024: 9390-9399.
[6] Yin Z, Xing E, Shen Z. Squeeze, recover and relabel: Dataset condensation at imagenet scale from a new perspective[J]. Advances in Neural Information Processing Systems, 2023, 36: 73582-73603.

**Questions:**

Please see weaknesses.

The quality of this paper can be highly improved if all the experiments and related works are compared and discussion properly.

---

### Official Review · Reviewer_MBhF · 2025-11-01

**Soundness:** 3
**Presentation:** 3
**Contribution:** 2
**Rating:** 2
**Confidence:** 4

**Summary:**

This paper introduces a novel task, Semi-Supervised Dataset Condensation (SSDC), and proposes a novel method named Semi-Supervised Dual Consistency Trajectory Matching (SSD). The goal is to synthesize a small, labeled dataset from a mixture of labeled and unlabeled samples, enabling efficient supervised learning without relying on fully labeled data. Particularly, a teacher model and a student model are employed. The teacher model is trained on both labeled and unlabeled data using SSL to generate reliable pseudo-labels. The Student model aims to achieve teacher-student agreement and meanwhile enhancing its robustness under perturbations. As a result, the synthetic dataset is optimized via trajectory matching between the student’s training dynamics on the real and synthetic data. Extensive experiments have shown effective performance in distilling a small dataset for supervised training.

**Strengths:**

- The paper introduces semi-supervised dataset condensation, which is a realistic topic that has lots of real-world applications.
- The proposed SSD elegantly combines semi-supervised learning with trajectory matching, which can effectively exploit unlabeled data meanwhile maintaining stability during training.
- The SSD consistently outperforms both supervised dataset condensation and self-supervised condensation baselines across multiple benchmarks and architectures.

**Weaknesses:**

- While the method is intuitively sound, the paper lacks formal theoretical analysis. It would be beneficial to demonstrate how dual consistency contributes to the convergence or stability of trajectory matching.
- The proposed method combines multiple modules, such as teacher–student training, consistency regularization, and trajectory matching. Computational efficiency on how these modules require and how they collaborate with each other is unclear. Moreover, the experiments are conducted under very small datasets, which remain unclear regarding their scalability to large datasets. Since the study aims to condense datasets, if there is no large-scale dataset experiments, the condensation would be unconvincing.
- Moreover, this paper does not examine how the ratio of labeled to unlabeled data affects condensation quality, nor whether unlabeled-only condensation is feasible within this framework.

**Questions:**

Please see the weaknesses.

---

### Official Review · Reviewer_WkBW · 2025-11-01

**Soundness:** 1
**Presentation:** 2
**Contribution:** 2
**Rating:** 2
**Confidence:** 4

**Summary:**

The paper extends trajectory-matching dataset condensation (MTT) to a semi-supervised setting: a teacher trained with unlabeled data produces pseudo-labels; a student is trained with dual consistency (teacher–student and augmentation consistency); the synthetic set is optimized by matching the student’s training trajectory on real vs. synthetic data. The paper claims improved accuracy and efficiency on small image classification benchmarks.

**Strengths:**

1. Clear motivation to exploit abundant unlabeled data in condensation.
2. The dual-consistency idea is simple and reasonable for handling noisy pseudo-labels.
3. Writing and figures make the pipeline easy to follow.

**Weaknesses:**

1. The method mainly combines known pieces (MTT-style trajectory matching + standard semi-supervised ingredients) and reads as an adaptation rather than a new principle.
2. Practical value for broader tasks is unclear if each task would require its own distilled set.
3. No result shows one fixed distilled set transferring across different architectures trained from scratch (cf. cross-architecture evaluations reported in DSA and M3D).
4. No wall-clock time, hardware/memory, or timing comparisons are reported.
5. Scale is small. Results are limited to MNIST/Fashion-MNIST/CIFAR-10.
6. Table 1 shows M3D not always > DM (e.g., CIFAR-10); setup alignment is unclear.
7. Minor presentation issues. Typos (e.g., 408 “12.2These results”).
The experimental support feels too light; closing the gap (fixed-set cross-architecture transfer, larger-scale data, and efficiency numbers) would likely require substantial additional work, so my current rating is 2 (reject).

**Questions:**

1. Is the distilled set intended for one task and one model family? What kind of transfer should readers expect across tasks and architectures?
2. Under a comparable compute/data budget, why prefer synthesizing a dataset over directly fine-tuning/KD of the student? If the distilled set is student-specific, the conceptual benefit beyond model-centric adaptation is not evident.
3. Could you comment on why a fixed distilled set is not evaluated across different architectures trained from scratch? If the set is student-specific, what is the intended use case?
4. Given the small-scale benchmarks, how should readers assess the claim that the synthesized dataset can stand in for a large-scale dataset?
5. What explains the M3D vs. DM numbers (e.g., CIFAR-10 at higher IPC) relative to M3D’s usual gains—are there configuration or protocol differences the reader should be aware of?

---

### Note · Authors · 2025-11-14

**Comment:**

We respectfully request to withdraw this submission due to the need to address methodological limitations identified during review. We appreciate the reviewers' efforts and feedback, which will guide our future revisions. Thank you for your understanding.

**Withdrawal Confirmation:**

I have read and agree with the venue's withdrawal policy on behalf of myself and my co-authors.